# Individual Expression of Hepatitis A Virus 3C Protease Induces Ferroptosis in Human Cells In Vitro

**DOI:** 10.3390/ijms22157906

**Published:** 2021-07-23

**Authors:** Alexey A. Komissarov, Maria A. Karaseva, Marina P. Roschina, Andrey V. Shubin, Nataliya A. Lunina, Sergey V. Kostrov, Ilya V. Demidyuk

**Affiliations:** Institute of Molecular Genetics of National Research Centre “Kurchatov Institute”, 123182 Moscow, Russia; mori-k@yandex.ru (M.A.K.); rochina78@yandex.ru (M.P.R.); andrey.v.shubin@gmail.com (A.V.S.); lunina@img.ras.ru (N.A.L.); kostrov@img.ras.ru (S.V.K.)

**Keywords:** hepatitis A virus 3C protease, regulated cell death, ferroptosis, transfection, transient expression, flow cytometry

## Abstract

Regulated cell death (RCD) is a fundamental process common to nearly all living beings and essential for the development and tissue homeostasis in animals and humans. A wide range of molecules can induce RCD, including a number of viral proteolytic enzymes. To date, numerous data indicate that picornaviral 3C proteases can induce RCD. In most reported cases, these proteases induce classical caspase-dependent apoptosis. In contrast, the human hepatitis A virus 3C protease (3Cpro) has recently been shown to cause caspase-independent cell death accompanied by previously undescribed features. Here, we expressed 3Cpro in HEK293, HeLa, and A549 human cell lines to characterize 3Cpro-induced cell death morphologically and biochemically using flow cytometry and fluorescence microscopy. We found that dead cells demonstrated necrosis-like morphological changes including permeabilization of the plasma membrane, loss of mitochondrial potential, as well as mitochondria and nuclei swelling. Additionally, we showed that 3Cpro-induced cell death was efficiently blocked by ferroptosis inhibitors and was accompanied by intense lipid peroxidation. Taken together, these results indicate that 3Cpro induces ferroptosis upon its individual expression in human cells. This is the first demonstration that a proteolytic enzyme can induce ferroptosis, the recently discovered and actively studied type of RCD.

## 1. Introduction

Regulated cell death (RCD) is a fundamental physiological process common to nearly all living beings [1]. In animals and humans, RCD is crucial for the development and tissue homeostasis, while abnormal RCD causes a wide range of diseases [2]. The mechanisms of cell death have been studied for more than 50 years, and today, more than ten RCD types are discriminated according to their inducers, key meditators, and cell morphology during and after death [3].

A wide range of molecules can induce RCD including certain viral proteolytic enzymes. Similar to other virus proteins, viral proteases are multifunctional. In addition to the main function of processing viral proteins, they can cleave cellular proteins. This can inhibit the transcription of cellular genes and cellular mRNA translation, block nuclear transport, and suppress the antiviral immune response [4,5]. Acting on cellular proteins, viral proteases can trigger RCD.

To date, numerous data indicate the ability of picornaviral 3C proteases to induce RCD. In most reported cases, these proteases induce classical caspase-dependent apoptosis [6,7,8]. In contrast, the rhinovirus 3C protease induces caspase-independent cell death; however, the underlying mechanism remains unclear [9]. Previously, we demonstrated that hepatitis A virus 3C protease (3Cpro) induces cell death that is also independent of caspase activity and features a unique cell morphology, specifically, the accumulation of cytoplasmic vacuoles with previously undocumented features. This suggested that the 3Cpro-induced cell death proceeds by an unknown pathway or is a new variant of a known RCD type [10,11,12].

Here, we analyzed the effect of 3Cpro on human cells in more detail. We found that cells died due to 3Cpro expression demonstrated necrosis-like morphological changes including permeabilization of the plasma membrane, loss of mitochondrial potential, partial chromatin condensation, as well as mitochondrial and nuclear swelling. Additionally, we showed that 3Cpro-induced cell death was efficiently blocked by ferroptosis inhibitors and was accompanied by intense lipid peroxidation. Taken together, these results show that 3Cpro induces ferroptosis in contrast to other picornaviral 3C proteases which commonly induce apoptosis. These data may indicate the presence of a new previously undefined function of 3Cpro or the new aspect of the interaction between the virus and host cell. In addition, ferroptosis is a recently discovered and actively studied RCD pathway. Although the molecular mechanism of this cell death type has been studied in quite detail, its biological functions are still to be understood. To date, all known specific ferroptosis inducers were represented by low molecular compounds. In this respect, this study is the first demonstration that proteolytic enzyme can induce this type of RCD. In this context, the results of the current study may become the basis for a new direction of research on the biological functions of ferroptosis.

## 2. Results

### 2.1. Ectopic Expression of 3Cpro and Its Inactive Form 3Cmut in Human Cells

Two genetic constructs were derived from the pCI vector: pCI-3C for the constitutive expression of active hepatitis A virus 3C protease (3Cpro) and pCI-3Cmut expressing the mutant enzyme with no proteolytic activity due to the Cys172-Ala substitution in the active site (3Cmut) (Figure 1A) [13,14]. HEK293, HeLa, and A549 cells were transfected with these genetic constructs, and 15 h post transfection (p.t.), the expression of the corresponding genes was confirmed by immunoblotting with antibodies against 3Cmut (Figure 1B). The 3Cpro production in HEK293 cells was higher compared to the other cell lines, since the 3Cpro level detected by immunoblotting [15] in the lysate of a smaller number of cells was comparable to that in HeLa and A549 cells.

### 2.2. Cpro Expression Induces Caspase-Independent Cell Death with Cytoplasmic Vacuolization

The cytotoxic effect after the transfection with pCI-3C/pCI-3Cmut was analyzed. The expression of mutant protease 3Cmut induced no cytotoxic effect (Figure 2A; solid lines). At the same time, the proportion of living cells in cultures expressing active 3Cpro decreased 15 h p.t. to reach the minimum 18 h p.t. (about 25% for HEK293 and less than 5% for HeLa and A549) and remained stable up to the end of the observation period (24 h p.t.) (Figure 2A; dashed lines).

The involvement of caspases in the 3Cpro-induced cell death was evaluated using the fluorescent caspase inhibitor FITC-VAD-fmk (Figure 2B). The proportion of cells with active caspases was about 15% after the transfection with either pCI-3C or pCI-3Cmut as demonstrated by flow cytometry (Figure 2C). At the same time, a considerable fraction of control cells treated with staurosporine (STS, a protein kinase C inhibitor, a well characterized inductor of caspase-dependent apoptosis [16]), showed the activation of caspases, which demonstrates that all the cell lines used are prone to caspase-dependent apoptosis. Thus, the data obtained confirm that the cytotoxic effect of 3Cpro depends on the proteolytic activity and the cell death is not accompanied by the activation of caspases.

We have also confirmed that 3Cpro-induced cell death is accompanied by cytoplasmic vacuolization as previously demonstrated [11]. Thus, a considerable fraction of HEK293 cells co-transfected with pCI-3C/pCI-3Cmut and pCI-EGFP (expressing the enhanced green fluorescent protein) showed green fluorescence 24 h p.t. as well as cytoplasmic vacuolization (Figure 2D; right). Nearly no cells were demonstrating green fluorescence 48 h p.t. At the same time, no cytoplasmic vacuolization was observed after co-transfection with pCI-3Cmut and pCI-EGFP, and cells remained attached to the substrate and emitted green fluorescence up to the end of the observation period (72 h p.t.) (Figure 2D; left). In the case of HeLa and A549, most cells transfected with pCI-3C/pCI-EGFP died 24 h p.t., and individual survived cells demonstrated green fluorescence but no cytoplasmic vacuolization. The data obtained likely indicate a higher susceptibility of HeLa and A549 cells to 3Cpro-induced cell death compared to HEK293. However, these data do not allow concluding about the cytoplasmic vacuolization in HeLa and A549 cells, since the vacuoles can be visualized only in EGFP-contrasted cytoplasm, while cells seem to die before they accumulate sufficient quantity of EGFP.

Thus, the effect of 3Cpro on human cells in the pCI-based expression system in vitro is similar to that previously reported by us [10,11].

### 2.3. Cells Expressing 3Cpro Acquire Necrotic Morphology and Are Characterized by Nuclei and Mitochondria Swelling

The morphology of HEK293, HeLa, and A549 cells transfected with pCI-3C or pCI-3Cmut was analyzed by staining with 1,1′,3,3,3′,3′-hexamethylindodicarbo-cyanine iodide (DiIC1(5)) and propidium iodide (PI) at different times p.t. to evaluate the mitochondrial metabolic activity and the plasma membrane integrity, respectively (Figure 3A). The vast majority of the cells expressing inactive 3Cmut at all time points had active mitochondria and intact plasma membrane, which are indicative of living cells (Figure 3B; 3Cmut). As active 3Cpro was expressed in culture, the proportion of living cells gradually decreased, and the proportion of cells with functionally inactive mitochondria and disrupted plasma membrane (i.e., with necrotic morphology) proportionally increased; at the same time, the proportions of other cell populations remained largely unaltered (Figure 3B; 3Cpro).

The morphology of nuclei and mitochondria in the 3Cpro-expressing cells was analyzed using fluorescence microscopy (representative pictures are presented for HeLa cells in Figure 4). For this purpose, DNA was stained with Hoechst 33342. Since the results of the experiment shown in Figure 3 indicated that 3Cpro-expressing cells lose mitochondrial membrane potential, mitochondria were visualized by immunostaining with anti-AIF and fluorescently labeled antibodies. Cells expressing inactive 3Cmut demonstrated normal nuclear and mitochondrial morphology (Figure 4, 3Cmut), whereas those expressing 3Cpro demonstrated partial chromatin condensation, as well as hypertrophy and rounding of their nuclei and mitochondria, indicating their swelling (Figure 4, 3Cpro).

### 2.4. 3Cpro-Induced Cell Death Is Effectively Blocked by Ferroptosis Inhibitors and Is Accompinied by Lipid Peroxidation

The necrotic morphology acquired by 3Cpro-expressing cells is typical of several RCD types: necroptosis, parthanatos, MPT-associated death, and ferroptosis. We examined the effect of their inhibitors on the cytotoxic effect of 3Cpro by analyzing the mitochondrial potential in transfected cell cultures, which is correlated with plasma membrane integrity according to the results shown in Figure 3. The presence of necrostatin-1 (Nec1, necroptosis inhibitor [17]), PJ34 (parthanatos inhibitor [18]), and cyclosporin A (CsA, inhibitor of MPT-associated death [19]) had no effect on the survival of 3Cpro-expressing cells (Figure 5A–C, Nec1, PJ34, and CsA). On the contrary, lipophilic antioxidant ferrostatin-1 (Fer1) and iron chelator desferrioxamine (DFO), common ferroptosis inhibitors [20,21,22], efficiently blocked the death of cells transfected with pCI-3C (Figure 5A–C, Fer1 and DFO), as well as those exposed to 50 µM erastin (Figure 5D)—the well-known ferroptosis inducer [23,24].

Lipid peroxidation is one of the key markers of ferroptosis. We analyzed this process in cells using BODIPY 581⁄591 C11 reagent which fluorescence, upon oxidation, shifts from red to green, providing a robust indication of lipid peroxidation. We found that both non-transfected cells and those transfected with pCI-3Cmut demonstrated predominantly red fluorescence, thus indicating the preponderance of the reduced form of BODIPY (Figure 6, Ctrl and 3Cmut, respectively). At the same time, cells expressing 3Cpro were characterized by the significant shift toward green fluorescence similar to the cells with induced lipid peroxidation due to cumene hydroperoxide exposure. These data indicate that 3Cpro-induced cell death is accompanied by intense lipid peroxidation.

Taken together, the results obtained indicate that 3Cpro-induced cell death is caused by active lipid peroxidation, and iron ions contribute to its progression. In turn, this allows us to conclude that the 3Cpro-induced cell death represents a form of ferroptosis.

## 3. Discussion

We recently demonstrated that human hepatitis A virus 3C protease (3Cpro) induces cell death with previously undescribed morphological features. The aim of this study was to find out whether the 3Cpro-induced cell death is a form of one of the known RCD types. Thus, 3Cpro and its inactive variant 3Cmut were expressed in several model cell lines. Fully compliant with the previous data [10,11], here, we showed that the observed cytotoxic effect of 3Cpro expression depends on the protease activity, and cell death is accompanied by cytoplasmic vacuolization without caspase activation.

We found that cells dying due to 3Cpro action demonstrated a necrotic morphology manifested as simultaneous plasma membrane disruption and loss of mitochondrial metabolic activity as well as nuclei and mitochondria swelling. This pattern is typical of several RCD types. We used characterized inhibitors of necroptosis, parthanatos, MPT-driven cell death, and ferroptosis to clarify whether the 3Cpro induced cell death represents a variant of these RCDs. Only ferroptosis inhibitors desferrioxamine (DFO) and ferrostatin-1 (Fer1) proved to effectively suppress cell death. Essentially, these inhibitors have different mechanisms of action. DFO is a Fe^2+^ chelator that has been shown to suppress ferroptosis by reducing of the intracellular labile iron pool content [25,26,27,28]. Fer1 is a lipophilic antioxidant capable of subverting ferroptosis through the specific inhibition of the lipid peroxidation [20,21,22,29]. Taken together, the data obtained suggest that 3Cpro induces ferroptosis mediated by Fe^2+^ ions and associated with the peroxidation of lipids. It is noteworthy that cells escaped 3Cpro-induced cell death under DFO concentrations higher than commonly used. We believe that this effect resulted from the difference in experimental systems used. Commonly, ferroptosis is induced using compounds added exogenously to the cells concurrently with or after DFO exposure. However, 3Cpro-induced ferroptosis is caused by the endogenously expressed enzyme, and DFO was added to the cells after the transfection to exclude its impact on transfection process. In this respect, it is not quite correct to compare directly DFO concentrations in previous studies and in our study. However, in contrast to DFO, Fer-1 efficiently blocked 3Cpro-induced cell death under commonly used concentrations. Our data are insufficient to clarify what features underlie this discrepancy in effective concentrations, but we believe that it originated from the relatively slow cell permeability of DFO [27,28]. In addition to the inhibitor analysis, we showed that cells expressing 3Cpro demonstrate intense lipid peroxidation, which is one of the key ferroptosis markers. Taken together, the results of the study indicate that the 3Cpro-induced cell death represents a form of ferroptosis.

In general, the morphology of the cells that died due to 3Cpro action is consistent with that of cells died through ferroptosis. Recent studies have shown that cells undergoing ferroptosis are usually characterized by necrosis-like morphological changes, including loss of plasma membrane integrity, cytoplasmic swelling (oncosis), swelling of cytoplasmic organelles, and moderate chromatin condensation (reviewed in [30]). All these features have been found in cells expressing 3Cpro both in the current and in previous studies [11].

Ferroptosis is a recently discovered RCD type which hallmark is an active membrane lipid oxidation [31]. The lipid oxidation is known to nonspecifically destabilize and disrupt the plasma membrane as well as mitochondrial, lysosomal, and EPR membranes (reviewed in detail in [30,32,33]). This could explain the consequent plasma membrane disruption and loss of mitochondrial activity observed during 3Cpro-induced cell death.

Concerning cytoplasmic vacuolization observed in 3Cpro-expressing cells, no such morphology has been described previously for ferroptosis. However, a similar morphology is observed in autophagic cells, and excessive autophagy and lysosome activity can sometimes promote ferroptosis [34,35]. Indeed, we have previously shown that vacuoles in the cytoplasm of the 3Cpro-expressing cells share several features with autophagosomes, but at the same time, these vacuoles are distinct from autophagosomes and have unique properties [11,36]. In addition, using common inhibitors of autophagy and vacuolization [37,38], we showed that 3Cpro-induced cell death does not depend on cytoplasmic vacuolization and autophagy induction [11]. Taken together, these data indicate that ferroptosis induction in 3Cpro-expressing cells is not promoted by autophagy.

It is worth noting another phenomenon revealed in the described 3Cpro-induced cell death. The proportion of cells expressing the target transgene post transfection is commonly used to evaluate the transfection agent efficiency; this parameter is often referred to as “transfection efficiency”. Our experiments with EGFP as a reporter indicated that the transfection efficiency of the used transfection agents does not usually exceed 40–50% for HEK293 cells and is much lower, ≈10–20%, for HeLa and A549 cells [13]. However, here, we have found out that the proportion of dead cells is much higher than the mean transfection efficiency after transient transfection (about 80% for HEK293 and over 95% for HeLa and A549). A similar effect was observed previously when 3Cpro was expressed in other genetic constructs [14]. The revealed distinctions can be attributed to the death of cells both expressing 3Cpro and neighboring ones. Such an effect was demonstrated for cells dying through ferroptosis, since ferroptosis occurring in one cell can reportedly spread to adjacent cells in a fast-propagating wave [39,40].

To date, numerous data indicate that different proteases can induce RCD; however, 3Cpro is the first reported proteolytic enzyme inducing ferroptosis. Moreover, all currently known specific ferroptosis inducers are synthetic low-molecular-weight compounds [33]. The found 3Cpro ability suggests this protease as a promising genetic ferroptosis-inducing agent, e.g., in cancer gene therapy. A wide range of cancer cells proved sensitive to ferroptosis; in particular, prostate, liver, lung, mammary gland, pancreas cancer as well as glioblastoma, acute myeloid leukemia, and diffuse B-cell lymphoma (reviewed in detail in [41,42]). Here, we also found that tumor HeLa and A549 cells are more susceptible to 3Cpro-induced ferroptosis than non-malignant HEK293 cells. This is confirmed by both higher intracellular 3Cpro levels and a higher proportion of living HEK293 cells compared to HeLa and A549. However, the data obtained are insufficient to conclude about the selective effect of 3Cpro on tumor cells. This requires further in vitro experiments with more cell lines and the subsequent confirmation in an in vivo system.

The data obtained are not enough to establish the biological role of 3Cpro as a ferroptosis inducer. It is common knowledge that the human hepatitis A virus has no direct cytopathic effect on hepatocytes, although the liver is the primary locus of virus replication. In vivo, the main factor that damages the liver is the death of infected hepatocytes mainly due to the activity of cytotoxic T cells and natural killer cells [43,44]. Moreover, the intracellular level of 3Cpro is apparently much lower during the infection compared to that in our experimental system. In this context, it is likely that 3Cpro affects certain cell substrates to maintain viral replication in vivo, while ferroptosis induction is a side effect of 3Cpro action. Apparently, low cellular levels of 3Cpro have no such side effect, while higher protease levels in our experimental system can induce it. A detailed analysis of the molecular mechanism of 3Cpro-induced cell death is needed to reveal the relationship between the ability of 3Cpro to induce ferroptosis and the viral life cycle. In the first place, the cellular targets of 3Cpro should be identified. This information can also extend our knowledge about the mechanism and biological role of ferroptosis.

## 4. Materials and Methods

### 4.1. Plasmid Expression Constructs

For pCI-3C construction, a DNA fragment containing the hepatitis A virus 3C protease (3Cpro) gene with EcoRI and KpnI sites was generated by PCR using the primers GACTGAATTCGCCACCATGTCAACTCTAGAAATAGCAGG and CAACGGTACCTTACTGACTTTCAATTTTCTTATCAATG (Evrogen, Moscow, Russia), and pBI-EGFP-3C [11] as the template. The fragment was purified using a Cleanup Standard kit (Evrogen, Moscow, Russia), digested with EcoRI and KpnI enzymes (SibEnzyme, Novosibirsk, Russia) and cloned into pCI (Promega, Madison, WI, USA) digested with the same enzymes. Plasmid pCI-3Cmut was constructed in the same way except that pBI-EGFP-3Cmut [11] was the source of the 3Cmut gene encoding inactivated 3Cpro with the Cys172-Ala substitution; pCI-EGFP was constructed previously [13]. All plasmids were amplified in *E. coli* TG1 cells and purified using a Plasmid Miniprep kit (Evrogen, Moscow, Russia).

### 4.2. Cell Cultures and Transfection

Human embryonic kidney HEK293, human cervical cancer HeLa (M-HeLa clone 11), and human adenocarcinomic alveolar basal epithelial A549 cell lines were obtained from the Russian Cell Culture Collection (St. Petersburg, Russia). The cells were cultured in DMEM/F-12 supplemented with 10% fetal bovine serum (ThermoFisher Scientific, Waltham, MA, USA) and 0.3 mg/mL glutamine (Paneco, Moscow, Russia) at 37 °C in a humidified atmosphere with 5% CO_2_ in 96- or 6-well plates (Corning, Corning, NY, USA) for 20–24 h until 60–80% confluence. For transfection, plasmid DNA-TurboFect and plasmid DNA-Lipofectamine 2000 (ThermoFisher Scientific, Waltham, MA, USA) complexes were prepared following the manufacturer’s protocol in serum-free OptiMEM (ThermoFisher Scientific, Waltham, MA, USA) and added to the wells; 4 h later, the medium was replaced with the fresh one or with the medium containing necrostatin-1 (5000, 500, 50, 5, or 0.5 μM), PJ34 (50, 5, 0.5, 0.05, or 0.005 μM), cyclosporin A (1000, 100, 10, 1, or 0.1 μM), ferrostatin-1 (200, 20, 2, 0.2, or 0.02 μM), or desferrioxamine (10,000, 1000, 100, 10, or 1 μM). For apoptosis and ferroptosis induction, non-transfected cells were incubated in the growth medium supplemented with 1 μM staurosporine (Sigma-Aldrich, Saint-Louis, MO, USA) or 50 μM erastin (Sigma-Aldrich, Saint-Louis, MO, USA), correspondingly, for 24 h prior to analysis.

### 4.3. Western Blotting

Whole-cell extracts obtained from 2∙10^5^ cells for HEK293 and 5∙10^5^ cells for Hela and A549 at 15 h post-transfection (p.t.) were separated by SDS-PAGE in 12% TGX Stain-Free gel (Bio-Rad, Hercules, CA, USA). After electrophoresis, the proteins were transferred to a nitrocellulose membrane (Bio-Rad, USA). The membranes were imaged using the stain-free system with a ChemiDoc MP imager (Bio-Rad, Hercules, CA, USA). Then, the membranes were blocked overnight with 5% nonfat dry milk (Bio-Rad, Hercules, CA, USA) in PBS (Paneco, Moscow, Russia) containing 0.1% Tween 20 (Sigma-Aldrich, Saint-Louis, MO, USA) and exposed to rabbit 3Cmut antiserum [14] diluted 1:300 overnight at room temperature. The membrane was routinely washed and incubated with the sheep anti-rabbit secondary antibodies (1 mg/mL) conjugated with horseradish peroxidase (Jackson ImmunoResearch Laboratories, Baltimore, PA, USA) for 2 h at room temperature. Immunoreactive bands were visualized with the Clarity Western ECL substrate (Bio-Rad, Hercules, CA, USA) following the manufacturer’s recommendations. The chemiluminescent signal was detected using a ChemiDoc MP imager.

The image analysis was carried out using the Image Lab software (Bio-Rad, Hercules, CA, USA). Total protein amount per track was evaluated following the manufacturer’s protocol for stain-free gels [15]. For each gel, a track with 1 ng of recombinant 3Cmut [14] served as a control.

### 4.4. Cell Viability Assay

Cells were transfected as described above in 96-well plates, and the viability of cells in transfected and non-transfected cultures was determined at the indicated time p.t. using a CellTiter 96 AQueous One Solution Cell Proliferation Assay kit (Promega, Madison, WI, USA). Briefly, the medium in 96-well plates was replaced with 100 μL of PBS, and 20 μL aliquots of the CellTiter 96 reagent were added per well. The absorbance was recorded at 490 nm with an Infinite M200 PRO microplate reader (Tecan, Maennedorf, Switzerland) immediately after the CellTiter 96 reagent was added and after 1 h of incubation at 37 °C in a humidified atmosphere with 5% CO_2_.

### 4.5. Lipid Peroxidation Assay

HEK293, HeLa, and A549 cells at 20 h p.t. with pCI-3C or pCI-3Cmut were incubated in growth medium supplemented with 10 µM BODIPY 581/591 C11 reagent (ThermoFisher Scientific, Waltham, MA, USA) for 30 min at 37 °C in a humidified atmosphere with 5% CO_2_. The same was done for non-transfected cells and cells pretreated with 100 µM cumene hydroperoxide for 2 h, which were used as negative and positive controls, respectively. After the incubation with BODIPY, cells were washed with growth medium, detached from the well bottom, and analyzed using flow cytometry.

### 4.6. Flow Cytometry

The medium in a 96-well plate was replaced at the indicated time points with 100 μL of PBS supplemented with 0.2 g/L EDTA and 50 μM FITC-VAD-fmk (Promega, Madison, WI, USA) or 50 μM 1,1′,3,3,3′,3′-hexamethylindodicarbo-cyanine iodide (DiIC1(5), ThermoFisher Scientific, Waltham, MA, USA), and the plate was incubated at 37 °C in a humidified atmosphere with 5% CO_2_ for 20 min. Then, the cells were stained with 0.3 µM propidium iodide (PI, Sigma-Aldrich, Saint-Louis, MO, USA) for 5 min and analyzed using an Accuri C6 flow cytometer (BD, Franklin Lakes, NJ, USA). The DiIC1(5) fluorescence was detected using 640 nm excitation and a 660–685 nm emission filter; PI, 488 and 600–620 nm; FITC-VAD-fmk, 488 nm and 518–548 nm. For the lipid peroxidation assay, fluorescence of the reduced form of BODIPY 581/591 C11 reagent was detected using 488 nm excitation and 565–605 nm emission filter, while fluorescence of the oxidized form was 488 and 518–548 nm. For each sample, at least 10,000 events corresponding to single cells by forward and side light scatter were acquired. The raw data were acquired using the Accuri C6 Software (BD, USA) and analyzed using the FlowJo software (BD, USA). The compensation matrix was calculated automatically by FlowJo using single-stained cells.

### 4.7. Fluorescence Microscopy

Cells were grown and co-transfected with pCI-3C/pCI-3Cmut and pCI-EGFP in a black 96-well plate with a transparent bottom (PerkinElmer, Waltham, MA, USA) and analyzed using an Axiovert 100 LSM510 META confocal microscope (Carl Zeiss, Oberkochen, Germany) 18 h p.t. For morphological analysis, the cells were transfected and 18 h later washed with PBS and fixed with methanol (Sigma-Aldrich, Saint-Louis, MO, USA) for 30 min on ice. The cells were washed twice with PBS, incubated for 30 min in blocking buffer (PBS, 5% fetal bovine serum, and 0.25% Triton X-100), and incubated for 20 h at 4 °C in the blocking buffer with goat IgG antibodies specific for the human AIF protein (Santa Cruz Biotechnology, Dallas, TX, USA) diluted 1:200. Next, the cells were washed twice with PBS and incubated in the blocking buffer containing donkey anti-goat IgG antibodies conjugated with Alexa Fluor 568 (ThermoFisher Scientific, Waltham, MA, USA) diluted 1:400 for 2 h at room temperature. The cells were washed twice with PBS, stained with 2 µg/mL Hoechst 33342 for 5 min, and analyzed under an Axiovert 100 LSM510 META confocal microscope (Carl Zeiss, Oberkochen, Germany).

### 4.8. Statistical Analysis

Comparison between unrelated groups was carried out using a nonparametric two-tailed Mann–Whitney U test. Differences were considered significant if the *p*-value was less than 0.05. Statistical analysis was performed using R (R Core Team (2020). R: A language and environment for statistical computing. R Foundation for Statistical Computing, Vienna, Austria; https://www.R-project.org/) (accessed on 15 May 2021) and RStudio version 1.3.1093 (RStudio Team (2019). RStudio: Integrated Development for R. RStudio, Inc., Boston, MA, USA; http://www.rstudio.com/) (accessed on 15 May 2021) software.

## Figures and Tables

**Figure 1 ijms-22-07906-f001:**
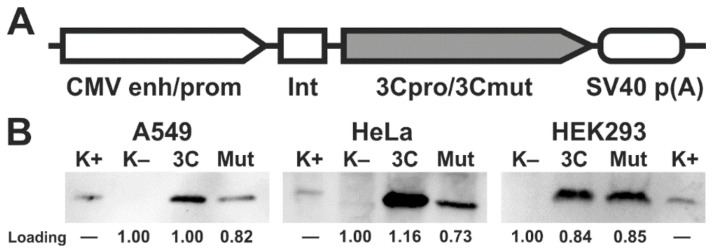
Expression of 3Cpro and its inactive form 3Cmut in human cells. (**A**) Expression cassettes in pCI-3C and pCI-3Cmut. CMV enh/prom, cytomegalovirus immediate-early enhancer/promoter; Int, chimeric human b-globin/IgG intron; 3Cpro/3Cmut, human hepatitis A virus 3C protease (intact or mutant, respectively) gene; SV40 p(A), late mRNA polyadenylation signal of SV40. (**B**) Analysis of 3Cpro and 3Cmut expression in HEK293, HeLa, and A549 cells by immunoblotting 15 h p.t.; 3C/Mut, lysate of cells transfected with pCI-3C/pCI-3Cmut; K–, lysate of non-transfected cells; K+, recombinant 3Cmut (1 ng). The lysates were prepared from 200,000 HEK293 or 500,000 HeLa/A549 cells. “Loading” shows the relative total protein per track estimated for each cell line using the stain-free technology.

**Figure 2 ijms-22-07906-f002:**
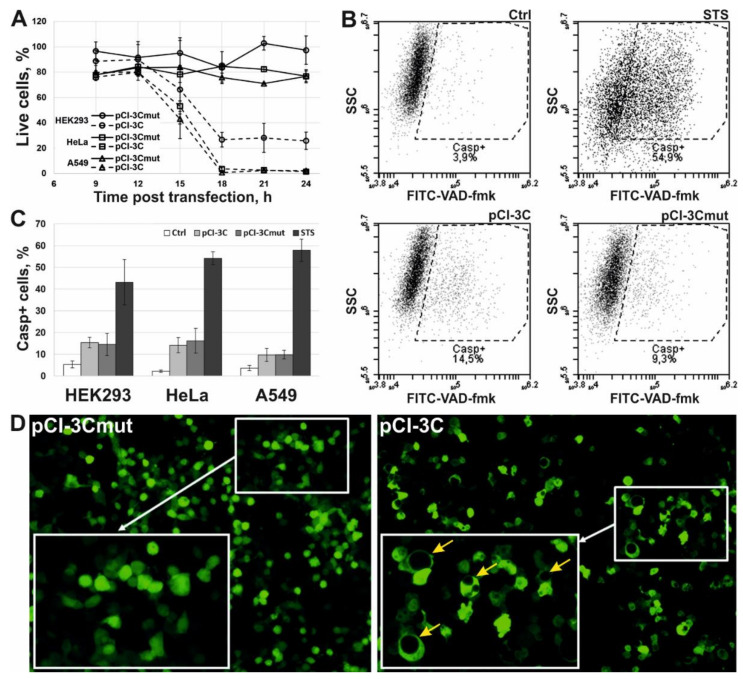
Effects of ectopic 3Cpro expression in HEK293, HeLa, and A549 cells. (**A**) Cytotoxic effect induced by 3Cpro expression. Results are expressed as the percentage of viable cells relative to non-transfected cells. (**B**) Representative flow cytometry images of non-transfected HeLa cells (Ctrl) and those transfected with pCI-3C and pCI-3Cmut, or incubated with 1 µM staurosporine (STS) and stained with FITC-conjugated caspase inhibitor (FITC-VAD-fmk). SSC, side scatter. (**C**) Caspase activation analysis using FITC-VAD-fmk in non-transfected cells (Ctrl) and those transfected with pCI-3C/pCI-3Cmut or incubated with STS. (**D**) Vacuolization of cells expressing 3Cpro. HEK293 cells were co-transfected with pCI-3Cmut (left) or pCI-3C (right) together with pCI-EGFP (1:1 by weight) and analyzed by fluorescence microscopy 24 h p.t. Yellow arrows indicate cells with vacuoles. All values are represented as mean ± SD of two independent experiments with triplicates (*n* = 6).

**Figure 3 ijms-22-07906-f003:**
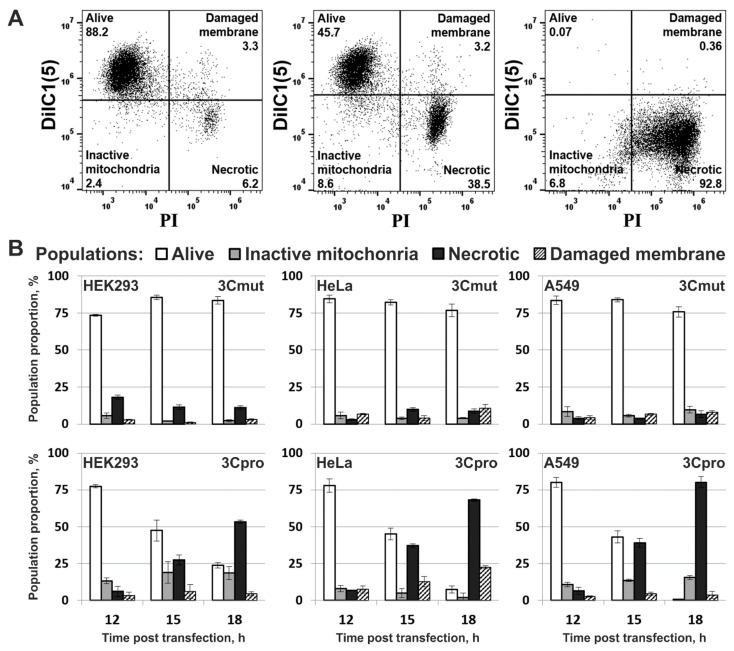
Flow cytometry analysis of morphology of 3Cpro expressing cells. (**A**) Representative dot plots of A549 cells stained with mitochondrial membrane potential sensitive dye 1,1′,3,3,3′,3′-hexamethylindodicarbo-cyanine iodide (DiIC1(5)) and propidium iodide (PI) 12 (left), 15 (middle), and 18 (right) h p.t. with pCI-3C. (**B**) Morphological changes in cell cultures expressing 3Cmut or 3Cpro. The proportions of different cell subpopulations discriminated on the basis of DiIC1(5) and PI staining are shown. All values are represented as mean ± SD of two independent experiments with triplicates (*n* = 6).

**Figure 4 ijms-22-07906-f004:**
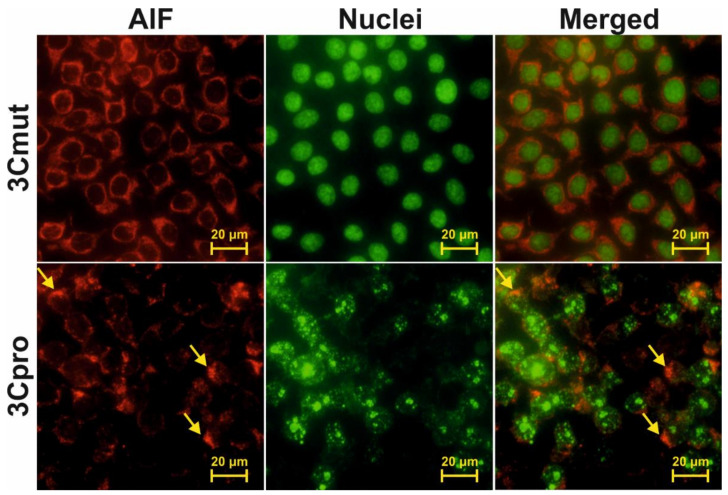
Analysis of nuclear and mitochondrial morphology in cells expressing 3Cpro. HeLa cells 18 h p.t. with pCI-3Cmut (3Cmut) or pCI-3C (3Cpro) were fixed and the mitochondrial protein AIF was visualized using AlexaFluor 568-labelled antibodies (AIF), while the nuclei were stained with Hoechst 33342 (Nuclei) as described in the Materials and Methods section. Arrows indicate aggregates of swollen mitochondria.

**Figure 5 ijms-22-07906-f005:**
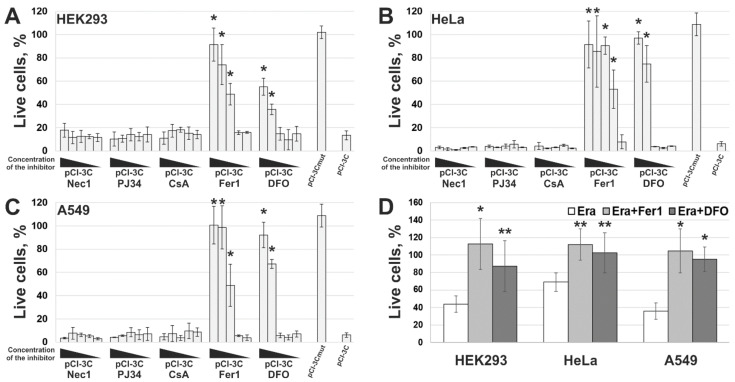
Effect of RCD inhibitors on 3Cpro-induced cell death. HEK293 (**A**), HeLa (**B**), and A549 (**C**) cells were transfected with pCI-3Cmut, or pCI-3C, or transfected with pCI-3C and exposed to different concentrations of necrostatin-1 (Nec1, 5000–0.5 μM), PJ34 (50–0.005 μM), cyclosporin A (CsA, 1000–0.1 μM), ferrostatin-1 (Fer1, 200–0.02 μM), and desferrioxamine (DFO, 10,000–1 μM). The viability of cells was analyzed 18 h p.t. as described in the Materials and Methods section, and the results were presented as the percentage of viable cells relatively to non-transfected cells. (**D**) Ferroptosis induction in the cell lines used. Cells were exposed to erastin (Era) alone or together with Fer1 (Era + Fer1) or DFO (Era + DFO). All values are represented as mean ± SD of two independent experiments with triplicates. Statistically significant differences between cells treated with indicated inhibitors and untreated cells (**A**–**C**), or between cells treated with Era alone and with Fer1/DFO (**D**) are marked with asterisks (Mann–Whitney U-test, *n* = 6, * *p* < 0.01, ** *p* < 0.05).

**Figure 6 ijms-22-07906-f006:**
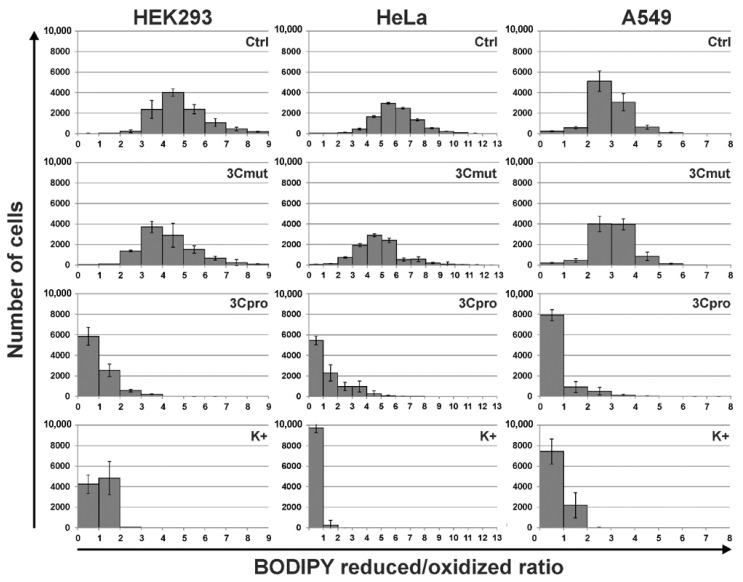
Lipid peroxidation upon 3Cpro/3Cmut expression. Non-transfected HEK293, HeLa, and A549 cells (Ctrl), transfected with pCI-3Cmut (3Cmut) or pCI-3C (3Cpro), and exposed to cumene hydroperoxide (K+) were incubated with BODIPY 581/591 C11 reagent, and analyzed using flow cytometry. For each cell in the population, the ratio of the fluorescence intensities of reduced (red) to oxidized (green) forms of BODIPY 581/591 C11 reagent were calculated. Values are represented as mean ± SD of two independent experiments with triplicates (*n* = 6).

## Data Availability

The data presented in this study are available in the article.

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
