# Peer review of "Individual Expression of Hepatitis A Virus 3C Protease Induces Ferroptosis in Human Cells In Vitro"

_ijms, 2021, doi:10.3390/ijms22157906_

Round 1

Reviewer 1 Report

In this manuscript, authors aim to describe a new role for the proteolytic enzyme 3Cpro in inducing ferroptotic cell death. Despite the results obtained, additional experimental data are required to say with certainty that is ferroptosis.

  1. Authors should improve the “Introduction” section with more information about the mayor morphological and biochemical features of ferroptosis, in order to drive the reader within the paper.

  1. Ferroptosis is a new type of regulated cell death (RCD), morphologically, genetically, and biochemically distinct from the other well-known cell death processes, such as apoptosis, necrosis, and autophagy. Indeed, unlike apoptosis or necrosis, ferroptosis does not display formation of apoptotic bodies, cell shrinkage and chromatin condensation, or swelling of the cytoplasm and organelles, and rupture of the cell membrane. In addition, unlike autophagy, ferroptosis does not display the formation of classical autophagosomes. Morphologically, ferroptotic cells exhibit ultrastructural changes in mitochondria such as outer mitochondrial membrane disruption and disappearance of the mitochondrial cristae. Furthermore, ballooning phenotype (i.e., the formation of a clear, rounded cell consisting mainly of empty cytosol, caused by cytoskeleton rearrangement, disruption of proteostasis, and membrane destabilization) can be used to microscopically recognize ferroptotic cells (DOI: 10.3390/cells9061505).

In section 2.3, authors affirm that “the proportion of cells with functionally inactive mitochondria and disrupted plasma membrane (i.e., with necrotic morphology) proportionally increased” (lines 121-123). In the same section they also assert that cells “expressing 3Cpro demonstrated partial chromatin condensation and their nuclei and mitochondria became hypertrophied and rounded, which indicates their swelling”.

This appears to conflict with the definition of cells undergoing ferroptosis. How can you explain it? Can you add TEM analysis to show mitochondrial ultrastructural changes and images of ballooning phenotype?

  1. Authors show that the treatment with both DFO and Ferrostatin-1 prevents 3Cpro cells mortality, suggesting that 3Cpro induces ferroptosis mediated by Fe2+ ions and oxidative stress. However, this is only a hypothesis. I strongly suggest measuring the LIP (labile iron pool) content of 3Cpro and 3Cmut cells at baseline levels and upon treatment with erastin alone and in combination with DFO/Ferrostatin-1.

  1. What staurosporine (section 2.2, line 72) is? Please, explain it.

  1. In order to demonstrate that the cell death mediated by 3Cpro is really ferroptosis, authors should perform some crucial experiments to evaluate:
  • lipid peroxidation rate (i.e., by using BODIPY staining);
  • mitochondrial membrane potential (i.e., by using TMRM staining);
  • cellular ROS (i.e., using DCF staining) and mitochondrial ROS (i.e., by using MitoSOX staining) amount.

The results should be also supported by western blot analysis of the pivotal player in the ferroptotic process (such as GPX4, iron metabolism proteins, VDAC etc.) or with other comparable experiments.

Reviewer 2 Report

In the manuscript, Alexey and colleagues claim that the human hepatitis A virus 3C protease causes ferroptosis in vitro. Although the topic is undoubtedly interesting, my major concern is whether ferroptosis indeed occurs and contributes to cell death because there is no substantial evidence except for the data using ferroptosis inhibitors. For example, the authors did not show any ferroptotic markers, such as increased levels of lipid peroxides and ferrous iron, and decreased levels of GSH and GPX4. In order to characterize the extent of ferroptosis, I have several comments that should be addressed by the authors. Specific comments are below.

Major points:

  1. Figure 5: The authors used a CellTiter assay for the validation of ferroptosis in cells. Since this is not a direct measurement of cell death, the authors should consider using a bona fide cell death assay such as propidium iodide (PI) staining like Figure 3.
  2. As mentioned above, I am concerned about the contribution of ferroptosis in the author’s model. As a positive control, the authors should verify whether 3C protease induces lipid peroxidation and impairs intracellular glutathione homeostasis. Also, since ferroptosis is an iron-dependent cell death mode, it would be necessary to show the changes in total iron contents (if possible catalytic ferrous iron). This is a minimum request.
  3. Figure 5: DFO, at concentrations higher than 1000 μM, rescued the cell. The concentration is very high. This point should be explained.
  4. The reviewer encourages the authors to confirm whether 3C protease-induced cell death is rescued by vitamin E, another ferroptosis suppressor.

Minor points:

  1. Figure 3: Representative images under microscopy should be shown for alive, damaged membrane, inactive mitochondria, and necrotic.
  2. Figure 4: AIF does not necessarily represent mitochondria. The authors should use MitoTracker as a mitochondrial marker.
  3. English should be carefully revised by a native English speaker or a professional editing service.

Reviewer 3 Report

Komissarov et al describe human hepatitis A virus 3C protease 14 (3Cpro) induced  cell death of huan cells in vitro. They demonstrate that the enzyme induces ferroptosis which is mediated by iron ions and cause oxidative stress. They state that this is the first demonstration that a proteolytic enzyme can induce ferroptosis. The paper is interesting but clearly needs revision.

  • Abstract: The abstract is not informative at all and clearly too short. Please largely expand and include clearly introduction, clear methods and results, clear, discussion and conclusions. Include facts such as experiments done, precise cells, results etc.
  • Introduction is too short. Please give a proper introduction to challenge interest. Explain why is important to know about the role of viral protease in ferroptosis. I have a hard time to believe that this is the first study showing regulated cell death by a viral proteolytic enzyme.
  • The entire paper need language editing by a native speaker.
  • Figure 2: It would be very helpful to have DIC images as well. Can you include?
  • Figure legends are very poor! What do we see hear? Please explain graphs y and x axis etc. etc.
  • Please include a series of experiments using a ROS scavenger.
  • Discussion: It would be very helpful to expand the discussion by looking beyond the box. How important are these results? What do they mean in terms of disease and potential clinical application?

Round 2

Reviewer 1 Report

The manuscript is now ready for publication.

Author Response

Together with the co-authors we thank the Reviewer for the positive assessment of our work.

Reviewer 2 Report

I am not satisfied with the revised manuscript because the authors did not address the most important points raised in the review process. The author’s responses are full of excuses. I previously suggested that the authors should show substantial evidence except for the data using ferroptosis inhibitors. Because ferroptosis is a mode of cell death, the authors should not have used a proliferation assay such as a CellTiter assay for the validation of ferroptosis. Also, since ferroptosis is an iron-dependent cell death mode, the results need the changes in total iron contents. I encouraged the authors to show any ferroptotic markers, such as increased levels of lipid peroxides, and decreased levels of GSH and GPX4. But the authors did not comply with any suggestion. I highly recommend reading and understanding a few of the latest articles and ones by Dr. Stockwell. Anyway, I believe this manuscript does not meet standard criteria of ferroptosis in its current state.

Reviewer 3 Report

The authors have improved the manuscript.

Round 3

Reviewer 2 Report

none.